# Evaluating the Impact of Probiotic Therapy on the Endocannabinoid System, Pain, Sleep and Fatigue: A Randomized, Double-Blind, Placebo-Controlled Trial in Dancers

**DOI:** 10.3390/ijms25115611

**Published:** 2024-05-21

**Authors:** Jakub Wiącek, Tomasz Podgórski, Krzysztof Kusy, Igor Łoniewski, Karolina Skonieczna-Żydecka, Joanna Karolkiewicz

**Affiliations:** 1Department of Food and Nutrition, Poznan University of Physical Education, 61-871 Poznan, Poland; 2Department of Biochemistry and Physiology, Poznan University of Physical Education, 61-871 Poznan, Poland; podgorski@awf.poznan.pl; 3Department of Athletics, Strength and Conditioning, Poznan University of Physical Education, 61-871 Poznan, Poland; kusy@awf.poznan.pl; 4Department of Biochemical Science, Pomeranian Medical University in Szczecin, Broniewskiego 24, 71-460 Szczecin, Poland; sanprobi@sanprobi.pl (I.Ł.); karolina.skonieczna.zydecka@pum.edu.pl (K.S.-Ż.)

**Keywords:** gut microbiome, probiotics, sports nutrition, supplementation, endocannabinoid system

## Abstract

Emerging research links the endocannabinoid system to gut microbiota, influencing nociception, mood, and immunity, yet the molecular interactions remain unclear. This study focused on the effects of probiotics on ECS markers—cannabinoid receptor type 2 (CB2) and fatty acid amide hydrolase (FAAH)—in dancers, a group selected due to their high exposure to physical and psychological stress. In a double-blind, placebo-controlled trial (ClinicalTrials.gov NCT05567653), 15 dancers were assigned to receive either a 12-week regimen of *Lactobacillus helveticus* Rosell-52 and *Bifidobacterium longum* Rosell-17 or a placebo (PLA: *n* = 10, PRO: *n* = 5). There were no significant changes in CB2 (probiotic: 0.55 to 0.29 ng/mL; placebo: 0.86 to 0.72 ng/mL) or FAAH levels (probiotic: 5.93 to 6.02 ng/mL; placebo: 6.46 to 6.94 ng/mL; *p* > 0.05). A trend toward improved sleep quality was observed in the probiotic group, while the placebo group showed a decline (PRO: from 1.4 to 1.0; PLA: from 0.8 to 1.2; *p* = 0.07841). No other differences were noted in assessed outcomes (pain and fatigue). Probiotic supplementation showed no significant impact on CB2 or FAAH levels, pain, or fatigue but suggested potential benefits for sleep quality, suggesting an area for further research.

## 1. Introduction

The endocannabinoid system (ECS) plays a critical role in modulating physiological processes such as nociception, mood, and immune responses. The ECS exerts its effects primarily through its bioactive lipids, anandamide (AEA; N-arachidonoylethanolamine) and 2-arachidonoylglycerol (2-AG), which interact with cannabinoid receptors 1 (CB1) and 2 (CB2). CB1 receptors are predominantly expressed in the nervous system, where they influence pain perception, mood, and appetite. Conversely, CB2 receptors are more commonly found on immune cells [1].

Unlike classical neurotransmitters, endocannabinoids such as anandamide are not stored in vesicles but are synthesized on demand. This synthesis is triggered by cellular demand and occurs in response to increases in intracellular calcium levels, which activate the biosynthetic enzymes [2]. Another distinctive feature of cannabinoids is their ability to mediate retrograde signaling. This process involves cannabinoids being produced post-synaptically and then traveling back across the synapse to bind to cannabinoid receptors on pre-synaptic neurons. This mechanism allows them to modulate neurotransmitter release, effectively regulating synaptic transmission in a feedback loop [3].

In addition to well-known endocannabinoids like anandamide (AEA) and 2-arachidonoylglycerol (2-AG), other lipid-based molecules such as palmityloylethanolamide (PEA) and palmitoylglycerol (PAG) also function as endocannabinoid-like substances. These compounds, structurally similar to classic endocannabinoids, do not bind directly to cannabinoid receptors. Instead, PEA enhances AEA activity through an “entourage effect,” offering anti-inflammatory and analgesic effects by inhibiting AEA breakdown and affecting non-cannabinoid pathways like peroxisome proliferator-activated receptors (PPAR-α) [4]. Additionally, these substances can interact with receptors such as the Transient Receptor Potential Vanilloid type-1 (TRPV1), a non-selective cation channel involved in pain and inflammation, which demonstrates the endocannabinoid system’s complex interactions with broader signaling networks [5].

Following its synthesis, the main endocannabinoid, anandamide, is primarily broken down by fatty acid amide hydrolase (FAAH), an enzyme that hydrolyzes anandamide into arachidonic acid and ethanolamine, thus regulating its availability and activity at cannabinoid receptors [6]. There has been substantial progress in understanding the biochemical and pharmacological characteristics of FAAH, including its structure and the interaction with inhibitors that could potentially treat conditions like inflammatory pain [7]. FAAH’s importance as a target for therapeutic intervention, especially for neuropsychiatric and neurological disorders like chronic pain, has led to innovative approaches in drug discovery (chemical graph mining, quantitative structure–activity relationship (QSAR) modeling, and molecular docking) to repurpose existing drugs as FAAH inhibitors [8].

Alongside the endocannabinoid system (ECS), the gut microbiota (GM) emerges as a critical factor capable of modulating immune functions and signaling pathways associated with daily well-being [9]. The complex ecosystem of the GM interacts with the body’s physiological processes through various mechanisms, including the production of neurotransmitters and the modulation of inflammatory responses. Studies have shown that changes in the composition and function of the GM can affect the central nervous system, thereby impacting emotional health, sleep patterns, and pain thresholds, suggesting a profound bi-directional communication within the gut–brain axis [10].

Regular exercise has been shown to increase the levels of endocannabinoids while also positively altering the composition of the GM [11,12]. However, overtraining and insufficient recovery can disrupt this beneficial balance, leading to dysregulation in both the ECS and gut microbiota. Such disturbances can manifest as increased inflammation, mood instability, and a weakened immune system, ultimately compromising physical and mental health [13,14].

The current experiment has been designed involving dancers, which is a group particularly vulnerable to significant physical and psychological stress due to their intensive training regimens. Psychological stress has been identified as a significant factor that increases the risk of injuries among dancers as observed in a study on first-year contemporary dance students [15]. Additionally, the habit of continuing to train despite experiencing pain, common among dancers, not only diminishes the quality of their movements but also exacerbates their psychological stress, further compounding their risk of injury. This behavior and its impacts on movement quality and stress levels were detailed in a separate study [16]. These dynamics highlight the necessity for a comprehensive approach to injury prevention and management tailored to dancers, who are exceptionally susceptible to both physical and psychological stress due to their demanding training schedules.

The primary outcomes of our study are the levels of cannabinoid receptor 2 (CB2) and fatty acid amide hydrolase (FAAH). We aimed to assess whether probiotic supplementation (Sanprobi Stress; *Lactobacillus helveticus* (Rosell-52) and *Bifidobacterium longum* (Rosell-17)) could reduce CB2 expression without affecting FAAH levels, suggesting a mechanism where the probiotic modulates the ECS by targeting CB2 receptors, which are involved mainly in immune responses. Alternatively, the probiotics could reduce inflammation through non-ECS pathways such as enhancing intestinal barrier integrity, modifying gut microbiota, or direct anti-inflammatory actions. If these mechanisms decrease FAAH activation, they would maintain higher levels of anandamide, which helps manage inflammation and pain. This could reduce systemic inflammation, potentially improving the well-being and performance of dancers under high stress.

The secondary outcomes of our study include indirect indicators that could reflect the functioning of the endocannabinoid system. These encompassed the threshold for mechanical pain perception, the occurrence of abdominal pain, assessments of sleep quality and sleep onset latency, and evaluations of fatigue. These metrics were chosen to provide a comprehensive view of the physiological changes potentially influenced by alterations in the endocannabinoid system following probiotic therapy.

## 2. Results

### 2.1. Anthropometric Profile, Dietary Composition

Due to exclusions and the meticulous selection of participants based on anthropometric measurements and activity level, the required group size calculated for the study was not met. Recruitment began in October 2022 and concluded in March 2023. In the experimental protocol, from an initial pool of 51 volunteers, a selection process was conducted to identify participants who met the inclusion criteria. This resulted in the selection of 26 female individuals. However, when it came time for the collection of blood and stool samples, only 20 dancers attended. Additionally, three of these participants did not return to collect the study’s supplement, and they did not provide reasons for their absence. Ultimately, 17 female dancers completed the study.

During the statistical analysis phase, two participants were excluded as outliers. This decision was based on the one participant’s body mass index (BMI), which indicated overweight status and was significantly different from the BMI values of the other group members. Due to substantial discrepancies in the FAAH and CB2 results of another participant, which were over two times higher than those observed in the rest of the group, this individual was also excluded from further statistical analysis of blood markers and questionnaires.

The flow of participants’ involvement in the study is presented in a diagram (Figure 1). During the course of the study, none of the participants reported any adverse effects or chose to withdraw due to such effects. This observation suggests that the interventions, including the probiotic supplementation and placebo treatments, were well tolerated among the participants.

All participants were female. Based on the provided baseline characteristics in Table 1, the placebo group (PLA) and the probiotic group (PRO) appear to be well-matched across several parameters. The age range is similar between the groups with a mean age of approximately 21 for PLA and 20 for PRO. Both groups have nearly identical body fat percentages and display no significant differences in body mass index (mean 21 in both groups). The physical activity levels and hand-grip test results are also comparable between the two groups, suggesting that the study participants were likely to be homogeneous in terms of physical characteristics and fitness levels at the outset of the study.

Diet is a significant factor that modifies the gut microbiome and potentially the endocannabinoid system; therefore, an analysis of dietary habits was conducted. The comparative analysis of the participants’ diets revealed no significant differences between the groups in energy and main nutrients. Both groups have overlapping ranges in their intake, suggesting there were no significant statistical differences in their diets. The estimated energy and fiber intakes shows some difference in mean values, yet the ranges still overlap, suggesting individual variation but no clear division between the groups. The characteristics of the diet are presented in Table 2.

### 2.2. Endocannabinoid System, Pain, Sleep and Fatigue

In Table 3, changes in levels of the cannabinoid receptor 2 (CB2) and fatty acid amide hydrolase (FAAH) before and after 3 months of supplementation with either probiotics (PRO) or placebo (PLA) are presented. To evaluate the expression of proteins in the endocannabinoid system, the concentrations of cannabinoid receptor 2 (CB2) and fatty acid amide hydrolase (FAAH) in the serum were measured using enzyme-linked immunosorbent assays (double-antibody sandwich ELISA).

Figure 2 displays the comparative results of CB2 and FAAH levels before and after supplementation in the PLA and PRO groups, including 95% confidence intervals.

In Table 4, the data reflect an indirect assessment of the endocannabinoid system’s activity following a three-month supplementation period with either probiotics or a placebo, encompassing several domains: pain, sleep and fatigue.

## 3. Discussion

### 3.1. Endocannabinoid System

To date, clinical trial results evaluating the levels of fatty acid amide hydrolase (FAAH) following probiotic supplementation have not been published. This represents an untapped area within clinical research that merits attention given the enzyme’s role in degrading endocannabinoids. Although studies on cannabinoid receptor 2 (CB2) are beginning to appear, the effects of probiotics on FAAH have not been extensively explored.

For the PLA group, the CB2 levels were measured at 0.86 ng/mL before supplementation and 0.72 ng/mL after supplementation. In the PRO group, CB2 levels were recorded at 0.55 ng/mL before supplementation and 0.29 ng/mL after 3 months. The two-way ANOVA showed an F-value (1, 13) of 0.64612 and a *p*-value of 0.43595, suggesting no significant effects for group interaction, time, or group-by-time interaction.

In a landmark study from the field it was discovered that the administration of *Lactobacillus acidophilus* in mice could upregulate the intestinal expression of cannabinoid receptors, significantly mitigating visceral pain through a mechanism mediated by CB2 and opioid receptors, thereby underscoring the endocannabinoid system’s (ECS) pivotal role in managing intestinal health [17]. Building on these and further insights, a recent comprehensive review elaborated on the ECS’s regulatory functions across gastrointestinal physiology. This review methodically detailed the system’s impact on gastrointestinal motility, secretion, inflammation, and its influence on visceral pain perception [18]. Inflammatory bowel disease (IBD), like Crohn’s disease and ulcerative colitis, follows irritable bowel syndrome as another group of chronic gastrointestinal conditions where the endocannabinoid system modulates inflammation and homeostasis, suggesting cannabinoid-based therapies as promising potential treatments [19].

IBS, along with migraines and fibromyalgia, is considered part of the hypothesized “endocannabinoid deficiency syndrome” as proposed by researchers. This theory suggests that low endocannabinoid levels might contribute to the symptoms of these conditions, such as chronic pain and digestive issues. Addressing endocannabinoid deficiencies could, therefore, offer a new avenue for treatment. This concept supports the potential benefits of cannabinoid-based therapies in managing these disorders [20].

In the first clinical study of its kind, 20 Caucasian women with mild to moderate abdominal pain received *Lactobacillus acidophilus* NCFM either alone or combined with B-LBi07 for 21 days. While L-NCFM alone was associated with increased mu-opioid receptor (MOR) activity, it also led to a decrease in cannabinoid receptor 2 (CB2) expression [21]. In a study from 2024, it was discovered that combining the prebiotic inulin with the probiotic *Lactobacillus rhamnosus* significantly increased the expression of the CB2 receptor while leaving CB1 receptor expression unchanged. This finding opens new avenues for exploring how dietary components can influence the endocannabinoid system, particularly in modulating specific receptor pathways without affecting others, in patients with coronary artery disease [22].

In dogs with colonic dysmotility, administration of the probiotic resulted in significant changes in the intestinal mucosa. Specifically, the treatment led to a decrease in the number of mucosal mast cells and an increase in cells expressing cannabinoid receptors CB1 and CB2, which are changes that correlated positively with improved clinical scores for canine chronic enteropathy activity index (CCECAI) post-treatment [23]. In a study examining the impact of antibiotic-induced intestinal dysbiosis on mice, significant shifts in gut microbiota were observed, including increases in *Bacteroides, Clostridium coccoides*, and *Lactobacillus* species, and a decrease in *Bifidobacterium*. This dysbiosis was linked to a notable upregulation of cannabinoid receptor 2 (CB2) expression. An enhanced expression of CB2 was associated with reduced visceral pain and increased colonic contractility, highlighting the role of commensal microbiota in modulating gut neuro-immune sensory systems and potentially informing treatments for conditions like irritable bowel syndrome (IBS) [24].

It was not possible for us to conduct chromatography, which is considered the most sensitive method. However, the detection of CB2 protein in blood is still feasible due to its expression on the surface of certain human peripheral blood leukocytes, specifically B cells. This is supported by findings showing CB2 protein localization at both extracellular and intracellular sites, enabling its measurement in blood samples [25].

For FAAH levels, the PLA group had levels of 6.46 ng/mL before supplementation and 6.94 ng/mL afterwards. In the PRO group, FAAH levels were 5.93 ng/mL before supplementation and 6.02 ng/mL after supplementation. These FAAH level measurements resulted in an F-value (1, 13) of 0.14627 and a *p*-value of 0.70831, reflecting no statistically significant differences between groups over time.

One of the earliest studies to investigate the interaction between probiotics and FAAH modulation demonstrated that inhibiting FAAH can alleviate depression-like symptoms in stressed rats. In this experiment, rats subjected to repeated social defeat were treated with the FAAH inhibitor URB694, which not only reduced depressive-like behaviors and neuroendocrine changes but also stabilized shifts in gut microbiota and lipid profiles [26]. In a molecular study, the effects of both active and heat-inactivated forms of *Akkermansia muciniphila*, along with its derived outer membrane vesicles (OMVs) and cell-free supernatant, were investigated on the gene expression related to the endocannabinoid system (ECS) in Caco-2 and HepG-2 cell lines. The study focused on key ECS components including cannabinoid receptors 1 and 2 (CB1 and CB2), fatty acid amide hydrolase (FAAH), and various peroxisome proliferator-activated receptors (PPARs α, β/δ, and γ). Quantitative real-time PCR analysis demonstrated a significant modulation of these genes, suggesting that both bacterial forms and their derivatives might influence metabolic pathways, offering potential therapeutic strategies against obesity, metabolic disorders, and liver diseases [27].

Some sources suggest that FAAH may be undetectable in blood. Nonetheless, measurements were conducted according to the manufacturer’s instructions for serum assays. Despite FAAH being an intracellular enzyme, there are two isoforms in humans: FAAH1, located on intracellular membranes, and FAAH2, which may be membrane-bound with a potentially extracellular orientation of the active site [28]. Additionally, studies on the detection of cell-free DNA in plasma indicate that physically active individuals, such as dancers, subjected to rigorous physical exertion and psychological stress, may experience cellular damage due to inflammatory responses [29,30]. These factors could potentially contribute to the presence of FAAH in the serum.

### 3.2. Pain, Sleep and Fatigue

Participants in the PLA group had abdominal pain scores of 4.1 before supplementation and 3.5 after supplementation with pressure-pain thresholds of 26.4 before supplementation and 22.5 after supplementation. Participants in the PRO group had abdominal pain scores of 3.1 before supplementation and 2.8 after supplementation with pressure-pain thresholds of 20.9 before supplementation and 19.9 after supplementation. The two-way ANOVA yielded an F-value (1, 13) of 0.52525 and a *p*-value of 0.48145 for abdominal pain. The analysis for pressure-pain thresholds reported an F-value (1, 13) of 0.32551 and a *p*-value of 0.57805.

While probiotics have shown some effectiveness in alleviating abdominal pain associated with irritable bowel syndrome (IBS) in children, results in older adult groups are not as conclusive. In our study, we did not observe significant differences between the probiotic and placebo groups, highlighting the variability in probiotic efficacy across different age groups and potentially different IBS symptoms [31,32,33].

Before our study, assessments of mechanical pain thresholds following probiotic administration were conducted in animal models by other researchers. These studies indicated that both diet-induced obese (DIO) mice and their normal weight (NW) counterparts showed significantly reduced sensitivity to mechanical stimulation when supplemented with a specific strain of probiotics [34,35].

For the PLA group, the sleep quality index scores were 0.8 before supplementation and 1.2 after supplementation. For the PRO group, these scores were 1.4 before supplementation and 1 after supplementation. The two-way ANOVA showed an F-value of 3.6491 and a *p*-value of 0.07841 (a trend toward quality increase). The sleep latency for the PLA group was 17.6 min before supplementation and 27.6 min after supplementation. For the PRO group, it was 15.2 min before supplementation and 15 min after supplementation. The two-way ANOVA reported an F-value of 1.5628 and a *p*-value of 0.23329.

A 2023 meta-analysis investigated the bidirectional relationship between gut microbiota and circadian rhythms, focusing on the impact of probiotic or prebiotic interventions on sleep quality and duration. Utilizing databases such as PubMed, Embase, CINAHL, and Web of Science, the study analyzed 18 articles from an initial pool of 219, concluding that gut microbiota modulation did not significantly improve sleep quality (*p* = 0.31) or duration (*p* = 0.43) [36]. In a 2024 clinical trial, it was observed that while the Pittsburgh Sleep Quality Index (PSQI) indicators of sleep quality did not significantly change after nine weeks of multi-strain probiotic (MSP) therapy, the time taken to fall asleep was reduced compared to placebo. The study, a randomized, double-blind, placebo-controlled trial, involved 70 healthy men and women who were supplemented daily with either multi-strain probiotic or a placebo [37]. In another clinical trial, 40 participants with stress-induced insomnia received either *B. breve* CCFM1025 or a placebo. Over four weeks, the probiotic group’s Pittsburgh Sleep Quality Index (PSQI) total scores significantly improved from 11.60 to 7.750, while the placebo group saw only minimal change from 10.10 to 8.650. This suggests *B. breve* CCFM1025 may effectively enhance sleep quality [38]. In a randomized, double-blind, placebo-controlled clinical trial involving 53 female participants with fibromyalgia syndrome (FMS), the effects of probiotics (4 × 10^10^ CFUs per day for 18 participants) and prebiotics (10 g of inulin per day for 17 participants) were compared over 8 weeks. The probiotics group showed significant improvements in Beck Depression Index, Beck Anxiety Index, PSQI, and pain (Visual Analogue Scale) scores compared to the placebo group, while the prebiotics group primarily saw significant improvements in subjective sleep quality [39].

In our study, dancers in the PRO group reported an activity level of approximately 16 h per week, and those in the PLA group reported about 17 h. These activity levels were maintained throughout the duration of the study, and participants were asked to report any changes in their activity levels. Epidemiological evidence suggests a positive correlation between the incidence of injury and the duration of dance training with a threshold identified at approximately 11.5 h per week. Gender-specific vulnerability is observed, wherein females display a heightened predisposition toward osseous injuries, whereas males exhibit a greater susceptibility to contusions and tendinopathies [40,41]. Fatigue levels for the PLA group were scored at 15.9 before supplementation and 13.1 after supplementation. In the PRO group, the scores were 18.6 before supplementation and 12.4 after supplementation. The two-way ANOVA showed an F-value (1,13) of 1.1803 and a *p*-value of 0.29702. A significant group interaction effect was observed in the assessment of fatigue; however, the reduction over time was similar across both groups. Active coping strategy scores for the PLA group were 3.2 before supplementation and 2.9 after supplementation. For the PRO group, these scores were 3.1 before supplementation and 2.8 after supplementation. The two-way ANOVA yielded an F-value (1, 13) of 0.00000 and a *p*-value of 1.0000. A recent systematic review highlighted that changes in the gut microbiota composition—particularly, reductions in pathogenic bacteria and increases in beneficial ones—are associated with diminished fatigue. These changes are thought to enhance gut barrier integrity, reduce inflammation, and adjust neurotransmitter levels that impact central nervous system function, thus influencing fatigue [42]. Research involving fecal microbiota transplantation (FMT) in patients with IBS has shown that altering the gut microbiota can inversely correlate with fatigue severity. Specifically, increases in beneficial bacteria such as *Alistipes* spp. and *Faecalibacterium prausnitzii* were associated with reduced IBS symptoms and fatigue [43]. Moreover, a pilot study with 70 patients suffering from post-infectious fatigue who received a multi-strain probiotic or placebo demonstrated that probiotics could significantly improve fatigue, mood, and quality of life with greater benefits observed in the probiotic group [44]. Additionally, a randomized controlled trial (RCT) highlighted that probiotics, particularly strains like *Lactobacillus casei* Shirota and *Bifidobacterium infantis* 35624, can reduce anxiety and inflammation in patients with chronic fatigue syndrome (CFS), further supporting the role of the gut microbiota in managing fatigue-related disorders [45]. Another study evaluated the impact of *Lactobacillus paracasei* ssp. paracasei F19, *Lactobacillus acidophilus* NCFB 1748, and *Bifidobacterium lactis* Bb12 on fatigue and physical activity in CFS patients. Initial findings indicated improvements in neurocognitive functions, although no significant changes were noted in fatigue or physical activity levels [46].

### 3.3. Limitations

Our study faced several limitations, including a sample size that was insufficient to attain the necessary statistical power. Additionally, the study lacked follow-up assessments which could have provided more comprehensive data over time. Diverse dosages and combinations, such as varying probiotic amounts or the inclusion of prebiotics, were not explored. Moreover, while immunoassay tests were utilized for measuring concentrations of endocannabinoid system indicators, chromatography would have been the preferred method for more precise and reliable results.

## 4. Conclusions

Although this study did not observe significant effects on cannabinoid receptor type 2 (CB2) and fatty acid amide hydrolase (FAAH) levels through probiotic supplementation, the findings suggest the need for further investigations with a larger cohort and varied dosages or the use of synbiotics to explore the full potential of probiotics on the ECS. Additionally, the trend toward improved sleep quality in the probiotic group highlights the possibility that enhancing sleep among athletes could be a valuable approach to optimizing recovery. These preliminary insights advocate for expanded research to refine probiotic formulations and dosing strategies to maximize their therapeutic benefits, particularly in athletic and high-stress environments.

## 5. Materials and Methods

### 5.1. Study Group

The calculations using G-Power application indicated that to have a strong chance (80% power) of detecting a medium-sized effect (Cohen’s d = 0.5) in our study, we would need 63 participants (male and female) in each group. This applies to the main variable in this study, which was CB2. The study included participants who were aged between 18 and 36 years and engaged in professional dancing activities with more than 8 h of training per week. Individuals were excluded from the study if they were younger than 18 or older than 36 years, had sustained injuries within the three months leading up to the study, had consumed prebiotics and/or probiotics in the three months preceding the study, had been hospitalized in the four weeks before the study began, had traveled to tropical countries within four weeks prior to the study, or had taken antibiotics, cannabis products, steroids, and anabolic steroids in the four weeks leading up to the study.

### 5.2. Study Protocol

Fifteen participants were selected for the analysis of primary and secondary outcomes with 10 receiving a placebo (PLA) and 5 receiving the probiotic (PRO) treatment. The randomization scheme was generated using a computer-based algorithm (block randomization with block size of four). The randomization was generated by the manufacturer of the probiotic under study. The distribution of participants between the placebo group (*n* = 10) and the treatment group (*n* = 5) in the final study group was impacted by elevated dropout rates and the presence of outliers, which were predominantly in the treatment group. While our objective was to sustain equal group sizes, we prioritized preserving the integrity of the randomization process.

Blinding included both the authors of the study and the participants. The randomized group of participants consumed a probiotic containing strains of *Lactobacillus helveticus* (Rosell-52) and *Bifidobacterium longum* (Rosell-17), at doses of 3 × 10^9^ CFU (colony-forming units) per capsule, in the mornings for 12 weeks (March–June, 2022), while another group received a placebo. Maltodextrin and cornstarch, which are carriers of the probiotic, were used as placebos in capsules identical to the probiotic in terms of mass and appearance. Supplemented probiotics were considered safe and well-tolerated in the healthy population, and the tested product held the necessary safety certificates. The research procedures were conducted twice: before the intervention and after its completion. The study was registered as a clinical trial in the clinicaltrials.gov database under the number NCT05567653. The study was conducted in accordance with the Helsinki Declaration and with the approval of the Bioethics Committee of the Poznan University of Medical Sciences (approval No. 412/22, 19 May 2022). To create the report, the CONSORT 2010 guidelines were used. No changes were made to the trial outcomes after the trial commenced.

### 5.3. Methods

In preparation for the clinical trial, detailed anthropometric measurements and dietary information were collected from the participants one week prior to the scheduled blood sampling. At the informational meeting, participants were instructed to maintain their usual dietary habits throughout the duration of the study while meticulously reporting any changes in diet, the occurrence of exclusion factors, and potential adverse effects. Additionally, they were advised to attend the blood sampling session in a rested state, after a light dinner, and following a 2–3 day break from training activities.

To evaluate the expression of proteins in the endocannabinoid system, the concentrations of cannabinoid receptor 2 (CB2) and fatty acid amide hydrolase (FAAH) in the serum were measured using enzyme-linked immunosorbent assays. FAAH and CB2 protein levels were measured using a double-antibody sandwich ELISA kits from SunRed Biotechnology Company. The assay range for FAAH was 0.15–30 ng/mL with a sensitivity of 0.116 ng/mL, and for CB2, it was 0.3–90 ng/mL with a sensitivity of 0.285 ng/mL. The procedure included negative control samples to ensure the specificity of detection for the proteins of interest. Blood samples (10 mL) were collected fasting, one day before the initiation of supplementation with either the probiotic or placebo, and then again after 3 months (12 weeks) from the start of the study. The blood samples were immediately frozen and stored until analysis ELISA could be conducted. A morphological analysis was also performed (flow cytometry) directly after the blood collection. The morphological analysis was conducted to exclude individuals with atypical results regarding specific blood cell populations, primarily leukocytes and lymphocytes, which are more specific to CB2 expression than other cells.

Nutritional status was assessed through analysis of dietary records and evaluation of blood morphology. Nutritional intake was assessed using the NUVERO software (2023 version, available for use in the browser, continuously updated), which was based on three-day dietary records. Supplementary to these tests, the pain threshold in response to mechanical pressure was assessed using an algometer applied to the thumb flexor, and grip strength was measured using a hand dynamometer. The mechanical pain threshold was tested through the three-time application of an algometer on the thumb adductor after adaptation to room temperature. The result is the average of three measurements. Grip strength was measured in a standing position with the arm directed downwards and the palm facing the thigh. The result is the average of three measurements.

Data were collected using measuring devices and the study leader’s computer (.xls format). Statistical analysis was performed using the STATISTICA 13.3 (TIBCO Software Inc., Palo Alto, CA, USA) package. For comparing baseline data between groups, the *t*-test or Mann–Whitney U-test was used if the normality condition was not met. The Shapiro–Wilk test was used to check the data for normality of distribution. To compare variables after 3 months of supplementation with either probiotic or placebo, the *t*-test or Wilcoxon test was used if the normality condition was not met. A two-way analysis of variance (ANOVA) for repeated measures was used to analyze the differences in the effect of time, group and time by group. For all statistical tests, a *p* value less than 0.05 was interpreted as statistically significant. For statistically significant results, the coefficients η^2^ are presented as an indicator of the effect size.

## Figures and Tables

**Figure 1 ijms-25-05611-f001:**
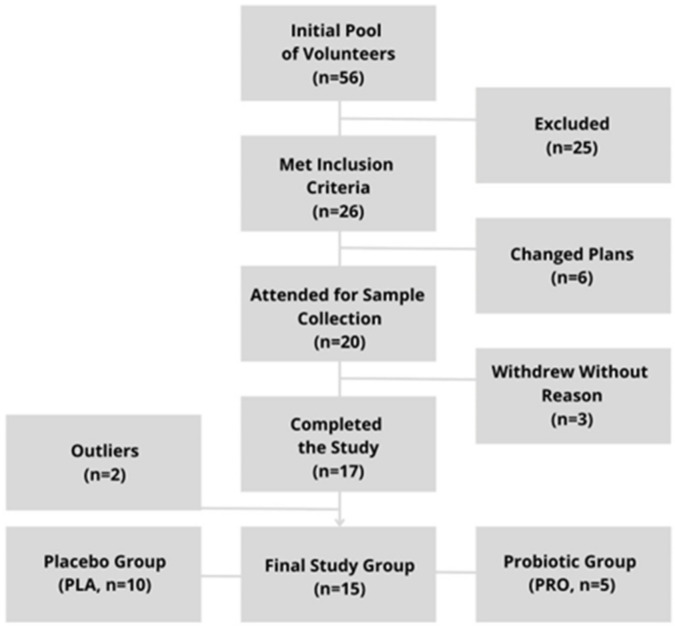
Study inclusion flow-diagram.

**Figure 2 ijms-25-05611-f002:**
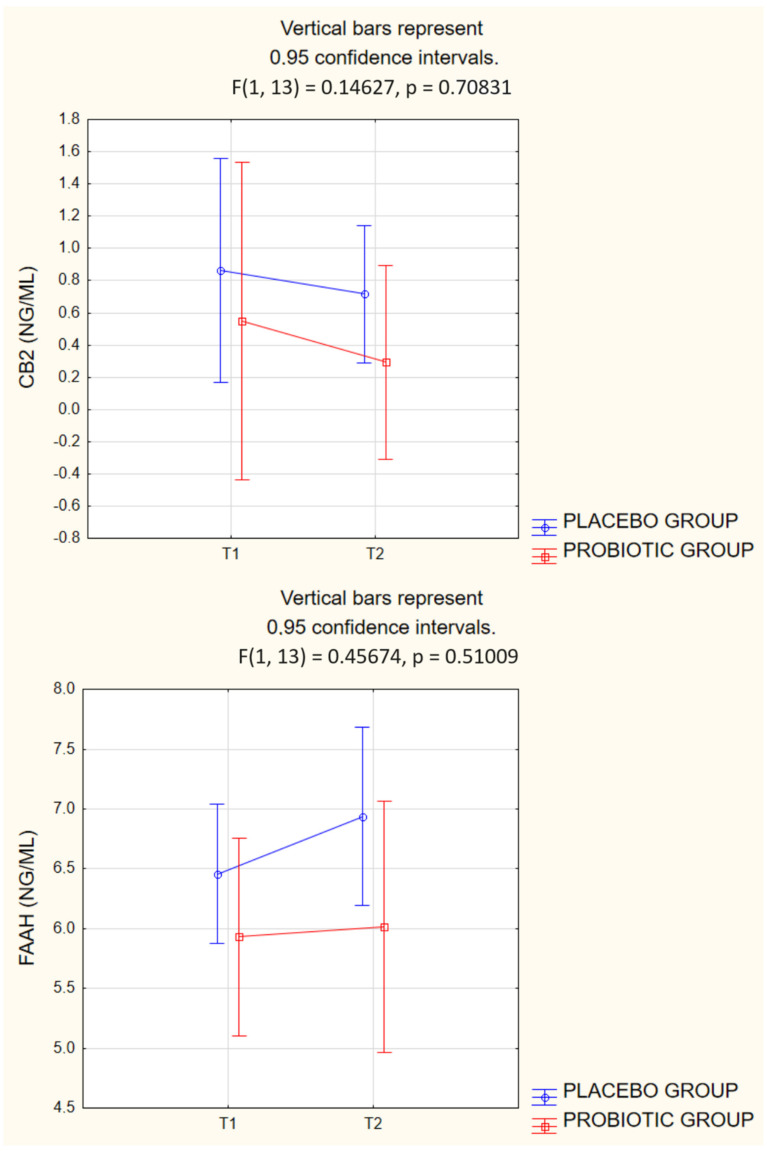
Graphs of two-way ANOVA for CB2 and FAAH levels in groups before and after probiotic therapy or placebo treatment.

**Table 1 ijms-25-05611-t001:** Baseline group characteristics.

	PLA (*n* = 11)	PRO (*n* = 5)	Total (*n* = 16)	PLA vs. PRO
	Mean ± SD (Min–Max)	Mean ± SD (Min–Max)	Mean ± SD (Min–Max)	*t*-Test/U Mann-Whitney (*p* Value)
Age [years]	20.55 ± 1.04 (19–22)	20.00 ± 1.30 (19–22)	20.44 ± 1.09 (19–22)	0.55
Body mass [kg]	58.07 ± 6.95 (49.40–68.70)	60.10 ± 7.31 (48.60–68.30)	58.08 ± 6.81 (48.60–68.70)	0.99
BMI (body mass index) [kg/m^2^]	21.05 ± 2.18 (17.70–23.40)	20.80 ± 2.29 (18.10–25.10)	21.02 ± 2.13 (17.70–25.10)	0.93
Fat [% body mass]	27 ± 4 (21–31)	27 ± 3 (25–31)	27 ± 3 (21–31)	0.84
Physical activity level [hours per week]	17 h 7 min ± 6 h 59 min (8 h–33 h)	16 h ± 9 h 46 min (9 h 30 min–29 h)	16 h 46 min ± 7 h 38 min (8 h–33 h)	0.69
Hand-grip test [kg]	28.13 ± 4.67 (20.07–38.13)	26.16 ± 5.33 (17.5–30.2)	27.51 ± 4.80 (17.5–38.13)	0.73

**Table 2 ijms-25-05611-t002:** Baseline diet characteristics of study participants.

	PLA (*n* = 11)	PRO (*n* = 5)	Total (*n* = 16)	PLA vs. PRO
	Mean ± SD (Min–Max)	Mean ± SD (Min–Max)	Mean ± SD (Min–Max)	*t*-Test/U Mann-Whitney (*p* Value)
Energy [kcal]	1999.23 ± 279.81 (1588.93–2578.45)	2325.54 ± 425.00 (1835.0–2842.6)	2101.20 ± 353.22 (1588.93–2842.6)	0.26
Protein [g]	85.29 ± 30.13 (49.52–154.26)	100.47 ± 21.09 (79.42–130.92)	90.03 ± 27.87 (49.52–154.26)	0.51
Fat [g]	74.56 ± 13.93 (57.69–95.74)	90.47 ± 18.72 (67.02–115.73)	79.54 ± 16.76 (57.69–115.73)	0.41
Carbohydrates [g]	271.12 ± 53.14 (194.89–359.42)	298.79 ± 57.98 (240.54–376.75)	279.77 ± 54.36 (194.89–376.75)	0.75
Fiber [g]	28.96 ± 15.13 (16.69–47.68)	21.36 ± 12.67 (15.03–27.63)	26.59 ± 14.44 (15.03–47.68)	0.91

**Table 3 ijms-25-05611-t003:** Markers related to endocannabinoid system before and after the 3 months of supplementation with probiotics or placebo (ELISA, serum—CB2, FAAH).

	PLA (*n* = 10) Mean ± SD (Min–Max)		PRO (*n* = 5) Mean ± SD (Min–Max)		2-Way ANOVA *p*-Value; (η^2^)		
	Pre	Post	Pre	Post	GROUP	TIME	GROUP × TIME
CB2 [ng/mL]	0.86 ± 1.21	0.72 ± 0.75	0.55 ± 0.32	0.29 ± 0.11	0.4168 (0.0513)	0.1865 (0.1302)	0.7083 (0.0111)
FAAH [ng/mL]	6.46 ± 0.68	6.94 ± 1.13	5.93 ± 1.15	6.02 ± 0.97	0.1293 (0.1679)	0.3500 (0.0674)	0.5100 (0.0341)

η^2^—effect size; *p* < 0.05 was considered as significant.

**Table 4 ijms-25-05611-t004:** Quality of life and behavioral indicators before and after three months of supplementation with probiotics or placebo.

	PLA (*n* = 10) Mean ± SD (Min–Max)		PRO (*n* = 5) Mean ± SD (Min–Max)		2-Way ANOVA *p*-Value; (η^2^)		
	Pre	Post	Pre	Post	GROUP	TIME	GROUP × TIME
Abdominal pain (ROME IV) [0 (never)–10 (always)]	4.1 ± 1.60	3.5 ± 1.43	3 ± 1.58	2.8 ± 1.79	0.2896 (0.0857)	0.1710 (0.1391)	0.4815 (0.0388)
Pressure–pain test [N]	26.4 ± 15.04	22.5 ± 7.04	20.9 ± 5.95	19.9 ± 5.46	0.4317 (0.0482)	0.3696 (0.0623)	0.5781 (0.0244)
Sleep quality (PSQI) [0 (best)–3 (worst)]	0.8 ± 0.63	1.2 ± 0.42	1.4 ± 0.55	1 ± 1.23	0.5365 (0.0301)	1.000 (0.0000)	0.0784 (0.2192)
Sleep latency (PSQI) [min]	17.6 ± 22.58	27.6 ± 36.24	15.2 ± 13.52	15 ± 10.60	0.5920 (0.0227)	0.2511 (0.1000)	0.2333 (0.1073)
Fatigue (FAS) (0–32)	15.9 ± 8.03	13.1 ± 5.41	18.6 ± 2.70	12.4 ± 4.62	0.7387 (0.0089)	0.0130 (0.3888)	0.2970 (0.0832)
Active strategies (Mini-COPE) [0 (never)–4 (always)]	3.2 ± 0.75	2.9 ± 0.74	3.1 ± 0.89	2.8 ± 1.10	0.8171 (0.0043)	0.0977 (0.1967)	1.0000 (0.0000)

η^2^—effect size; *p* < 0.05 was considered as significant.

## Data Availability

Data is contained within the article.

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
