# Peer review of "Evaluating the Impact of Probiotic Therapy on the Endocannabinoid System, Pain, Sleep and Fatigue: A Randomized, Double-Blind, Placebo-Controlled Trial in Dancers"

_ijms, 2024, doi:10.3390/ijms25115611_

Round 1

Reviewer 1 Report

Comments and Suggestions for Authors

The manuscript entitled “Evaluating the Impact of Probiotic Therapy on the Endocannabinoid System, Pain, Sleep and Fatigue: A Randomized, Double-Blind, Placebo-Controlled Trial in Dancers” is a small study describing effects of probiotics on dancers. The paper is easy to read, and the discussion is good, comprehensive, and relates their work to other work in the field well. Some issues should be addressed.

Rationale provided for dancers in the introduction, but not evident in the abstract. It is an unusual population for study so it might be good to provide that rationale earlier.

What was the rationale for providing placebo to 11 but only 5 treatments?

The graphs are difficult to read with small fonts and too many horizontal lines across the graphs.

In the results section it is not clear how CB2 and FAAH expression were evaluated. Was this protein or mRNA? What technique was used? Some details are provided in the materials and methods, but it should also be noted in the results, so the reader understands the data when presented. Related to this, is ELISA (presumably without any kind of protein isolation) an accepted measure of CB2 and FAAH? It certainly is not a measure of activation of the endocannabinoid system as stated on line 392; if anything, it is simply expression. Are the authors suggesting that these proteins are freely floating in the plasma or was there some kind of isolation performed? Especially since FAAH is intracellular, how do the authors account for detection? Were negative controls used to ensure they were actually detecting the proteins of interest or was there another approach used to verify they are actually detecting CB2 or FAAH? What was the morphology test used for?

In Table 4 title, the authors call these “indirect assessment of endocannabinoid system activation”. While this is likely true, this is more interpretation so should be limited to the discussion. The results table should simply be described as what these data were – perhaps “behavior”, “quality of life”, or another term that collectively describes these endpoints.

It is not entirely clear what contribution the paragraph about injury makes to the discussion (lines 296-304). Perhaps if this is in a different part of the discussion with more context as to its ties to the current data, that would help. Does this tie to the fatigue paragraph that follows?

Minor

Used “flow” twice in the sentence on line 129.

Line 205 need italics on “Lactobacillus acidophilus”; other places throughout the discussion need references to bacterial strains italicized – please check.

In the discussion there are several places in which a single sentence is a paragraph. Some of these ideas can be combined to make proper paragraphs throughout the discussion, especially near the end.

Author Response

Dear Reviewer,

Thank you very much for the detailed review. Below, I address the issues raised:

- Information about dancers being highly burdened by physical effort and psychological stress, which served as the basis for the hypothesis in studies of the endocannabinoid system, has been added to the abstract.

- Distribution of participants between the placebo group (n=11) and the treatment group (n=5) was influenced by higher dropout rates and the occurrence of outliers predominantly in the treatment group. Our goal was to maintain equal group sizes; however, we did not want to interfere with the randomization process since the studies will continue, and maintaining the integrity of this process is crucial for the validity of ongoing research.

- The issue with small fonts in the graphs has been addressed, and the new versions with improved readability have been attached (line 185).

- FAAH protein levels were measured using a double-antibody sandwich enzyme-linked immunosorbent assay (ELISA) kit. The assay range was 0.15-30 ng/ml, with a sensitivity of 0.116 ng/ml. This information has been added to the text to ensure that readers can fully understand the data when presented (line 174). As mentioned in section 3.3 on limitations, while ELISA is used here, chromatographic methods such as High-performance liquid chromatography–tandem mass spectrometry (HPLC–MS/MS) assays would be preferred for measuring FAAH levels in blood for increased accuracy [1]. The text has been revised to clearly indicate that the measurements pertain to protein expression, not activation of endocannabinoid system (line 443). No isolation was performed. The measurements were conducted according to the manufacturer's instructions for serum assays. This was the only available method in our research unit. As “Steve P.H. Alexander in xPharm: The Comprehensive Pharmacology Reference” states, in humans, two isoforms of FAAH exist: FAAH1, located on intracellular membranes, and FAAH2, which may be bound to membranes with a potentially extracellular orientation of the active site [2]. Additionally, studies on the detection of cell-free DNA in plasma indicate that physically active individuals, such as dancers, who are subjected to rigorous physical exertion and psychological stress, may experience cellular damage due to inflammatory responses [3, 4]. The procedure included negative control samples to ensure the specificity of detection for the proteins of interest. The morphological analysis was conducted to exclude individuals with atypical results regarding specific blood cell populations, primarily leukocytes, which are more specific to CB2 expression than other cells [5]. This helped ensure that the study's measurements were not influenced by unusual blood profiles.

- The title of Table 4 has been corrected to "Indicators Related to Quality of Life and Behavior" to more accurately describe the data presented (line 191).

- The paragraph about injury is part of the discussion related to fatigue. It highlights how training despite fatigue and pain can exacerbate issues with proper post-exercise recovery, tying directly into the broader theme of how physical stress impacts dancers.

- The sentence on line 129 has been revised to avoid the repetition of the word "flow."

- The reference to "Lactobacillus acidophilus" on line 205 and other mentions of bacterial strains throughout the discussion have been corrected to italics, ensuring consistent formatting in accordance with scientific naming conventions.

- The text in the discussion has been rearranged to occupy less space by combining single-sentence paragraphs into more substantial paragraphs, particularly towards the end, to enhance readability and coherence.

[1] Yapa, U., Prusakiewicz, J. J., Wrightstone, A. D., Christine, L. J., Palandra, J., Groeber, E., & Wittwer, A. J. (2012). High-performance liquid chromatography-tandem mass spectrometry assay of fatty acid amide hydrolase (FAAH) in blood: FAAH inhibition as clinical biomarker. Analytical biochemistry, 421(2), 556–565. https://doi.org/10.1016/j.ab.2011.10.042

[2] Steve P.H. Alexander, Fatty Acid Amide Hydrolase (FAAH), Editor(s): S.J. Enna, David B. Bylund, xPharm: The Comprehensive Pharmacology Reference, Elsevier, 2009, Pages 1-7, ISBN 9780080552323, https://doi.org/10.1016/B978-008055232-3.64098-X.

[3] Atamaniuk, J., Vidotto, C., Tschan, H., Bachl, N., Stuhlmeier, K. M., & Müller, M. M. (2004). Increased concentrations of cell-free plasma DNA after exhaustive exercise. Clinical chemistry, 50(9), 1668–1670. https://doi.org/10.1373/clinchem.2004.034553

[4] Fatouros, I. G., Destouni, A., Margonis, K., Jamurtas, A. Z., Vrettou, C., Kouretas, D., Mastorakos, G., Mitrakou, A., Taxildaris, K., Kanavakis, E., & Papassotiriou, I. (2006). Cell-free plasma DNA as a novel marker of aseptic inflammation severity related to exercise overtraining. Clinical chemistry, 52(9), 1820–1824. https://doi.org/10.1373/clinchem.2006.070417

[5] Castaneda, J. T., Harui, A., Kiertscher, S. M., Roth, J. D., & Roth, M. D. (2013). Differential expression of intracellular and extracellular CB(2) cannabinoid receptor protein by human peripheral blood leukocytes. Journal of neuroimmune pharmacology : the official journal of the Society on NeuroImmune Pharmacology, 8(1), 323–332. https://doi.org/10.1007/s11481-012-9430-8

Reviewer 2 Report

Comments and Suggestions for Authors

This research study investigated the role of some components of the endocannabinoid system after probiotic administration in a group of dancers.

Methods: 

- Provide more info on the ELISA test used.

- What is the placebo formulation ?

- Is the probiotic used already available on the market? 

Results:

- Table 4. Could be possible a placebo effect on the stress ? Placebo group displayed better sleep quality and decreased pain.

Discussion:

- the number of dancers in PRO group is too little to give some type of consideration.

Overall: the discussion is well written. Taken into account the limitations, this study could be better if more dancers will be recruited.

Comments on the Quality of English Language

English is fine.

Author Response

Dear Reviewer,

I greatly appreciate your thorough and insightful feedback. Below are my responses to the concerns you have highlighted:

- A more detailed description of the ELISA test is provided in line 174.

- Maltodextrin and cornstarch, which are carriers of the probiotic, were used as placebos in capsules identical to the probiotic in terms of mass and appearance. This information has been added to section 5.2. Study Protocol (line 424).

- Yes, the probiotic studied is commercially available in pharmacies under the name Sanprobi Stress.

- The limited number of participants in the PRO group was a consequence of stringent selection criteria, which included anthropometric measurements, levels of physical activity, grip strength, pain threshold, dietary composition, and hematological profiles. This approach was aimed at ensuring homogeneity within the study groups but resulted in a reduced sample size. Additionally, this research represents an emerging field of study, providing foundational data that will inform the design of more extensive future investigations.

Round 2

Reviewer 1 Report

Comments and Suggestions for Authors

While I appreciate that the authors addressed these concerns in the rebuttal, the information regarding participant numbers and details about the ELISA need to be in the paper.

Participant numbers and information about the negative controls and use of cellular morphology for the ELISA should go in the methods and the discussion regarding the ability to detect FAAH and CB2 in the plasma needs to go in the discussion.

Author Response

Dear Reviewer,

Thank you for pointing out the omission of this information in the text.

The information regarding the number of participants and the unequal distribution in the groups has been added in line 439.

The information about the use of negative controls for optimizing the procedure has been included in line 472.

The discussion on the extracellular detectability of FAAH has been included in line 272, and CB2 in 247.

The relevance of conducting a blood morphology analysis for leukocytes and lymphocytes has been addressed in line 485.

Reviewer 2 Report

Comments and Suggestions for Authors

Authors well answered to my comments and modified their manuscript accordingly.

Author Response

Dear Reviewer,

Thank you for reviewing the changes made after the initial round of reviews. In addition to the previously discussed modifications, the following updates have been made:

Information on the number of participants and the unequal group distribution has been added in line 439. Details on the use of negative controls for optimizing the procedure has been included in line 472. The discussion on the extracellular detectability of FAAH has been included in line 272, and CB2 in 247. The relevance of conducting a blood morphology analysis for leukocytes and lymphocytes has been addressed in line 485. The relevance of blood morphology analysis for leukocytes and lymphocytes has been addressed in line 485.